# Comparison of three dosing intervals for the primary vaccination of the SARS-CoV-2 mRNA Vaccine (BNT162b2) on magnitude, neutralization capacity and durability of the humoral immune response in health care workers: A prospective cohort study

**Darryl P. Leong**[1,2], **Ali Zhang**[3,4,5], **Jessica A. Breznik**[2,3,5,6], **Rumi Clare**[7], **Angela Huynh**[2], **Maha Mushtaha**[1], **Sumathy Rangarajan**[1], **Hannah Stacey**[3,4,5], **Paul Y. Kim**[2,8], **Mark Loeb**[2,9,10], **Judah A. Denburg**[2], **Dominik Mertz**[1,2,5,9,11], **Zain Chagla**[2], **Ishac Nazy**[7,12], **Matthew S. Miller**[3,4,5], **Dawn M. E. Bowdish**[2,3,5,13], **MyLinh Duong**[1,2,13]*

**1** Population Health Research Institute, McMaster University and Hamilton Health Sciences, Hamilton, Ontario, Canada, **2** Department of Medicine, Michael G. DeGroote School of Medicine, McMaster University, Hamilton, Ontario, Canada, **3** McMaster Immunology Research Centre, McMaster University, Hamilton, Ontario, Canada, **4** Department of Biochemistry & Biomedical Sciences, McMaster University, Hamilton, Ontario, Canada, **5** Michael G. DeGroote Institute for Infectious Disease Research, McMaster University, Hamilton, Ontario, Canada, **6** McMaster Institute for Research on Aging, McMaster University, Hamilton, Ontario, Canada, **7** McMaster Platelet Immunology Laboratory, McMaster University, Hamilton, Ontario, Canada, **8** Thrombosis and Atherosclerosis Research Institute, McMaster University, Hamilton, Ontario, Canada, **9** Department of Pathology and Molecular Medicine, McMaster University, Hamilton, Ontario, Canada, **10** Department of Epidemiology and Biostatistics, McMaster University, Hamilton, Ontario, Canada, **11** Department of Health Research Methods, Evidence, and Impact, McMaster University, Hamilton, Ontario, Canada, **12** McMaster Centre for Transfusion Research, McMaster University, Hamilton, Ontario, Canada, **13** Firestone Institute for Respiratory Health, The Research Institute of St. Joe's Hamilton, Hamilton, Ontario, Canada

* duongmy@mcmaster.ca

## Abstract

### Objectives

The dosing interval of a primary vaccination series can significantly impact on vaccine immunogenicity and efficacy. The current study compared 3 dosing intervals for the primary vaccination series of the BNT162b2 mRNA COVID-19 vaccine, on humoral immune response and durability against SARS-CoV-2 ancestral and Beta variants up to 9 months post immunization.

### Methods

Three groups of age- and sex-matched healthcare workers (HCW) who received 2 primary doses of BNT162b2 separated by 35-days, 35–42 days or >42-days were enrolled. Vaccine induced antibody titers at 3 weeks, 3 and 6–9 months post-second dose were assessed.

**Data Availability Statement:** The Population Health Research Institute (PHRI) is the sponsor of this STUDY. PHRI believes the dissemination of research results is vital and sharing of data is important. The minimum raw summary data underlying the results presented in this paper are available in the accompanying Supporting Information material. Individualized de-identified data cannot be publicly share, as it involves a small group of HCW in a small region; and indirect identifiers may risk identifying the study participants. Furthermore, consent for public disclosure of this information was not obtained and could pose a threat to confidentiality and violate privacy laws. Data may be disclosed upon request and approval of the proposed use of the data by a PHRI Review Committee. Requests for access to data may be sent to PHRI Publications Committee and the PHRI Contracts phri.contracts@phri.ca.

**Funding:** This study was supported by internal funding from the Population Health Research Institute, McMaster University and the Department of Medicine, Faculty Health Sciences. The funders had no role in the conception, design, data analysis, data interpretation or reporting of the study.

**Competing interests:** The authors have declared that no competing interests exist.

## Results

There were 309 age- and sex-matched HCW (mean age 43 [sd 13], 58% females) enrolled. Anti-SARS-CoV-2 binding (IgG, IgM, IgA) and neutralizing antibody titers showed significant waning in levels beyond 35 days post first dose. The second dose induced a significant rise in antibody titers, which peaked at 3 weeks and then declined at variable rates across groups. The magnitude, consistency and durability of response was greater for anti-Spike than anti-RBD antibodies; and for IgG than IgA or IgM. Compared to the shorter schedules, a longer interval of >42 days offered the highest binding and neutralizing antibody titers against SARS-CoV-2 ancestral and Beta (B1.351) variants beyond 3 months post-vaccination.

## Conclusions

This is the first comprehensive study to compare 3 dosing intervals for the primary vaccination of BNT162b2 mRNA COVID-19 vaccine implemented in the real world. These findings suggest that delaying the second dose beyond 42 days can potentiate and prolong the humoral response against ancestral and Beta variants of SARS-CoV-2 up to 9 months post-vaccination.

## Introduction

Most primary vaccination series employ an interval of 8–12 weeks between the prime and booster doses to promote optimal cellular and humoral immune responses. The BNT162b2 mRNA COVID-19 vaccine, has been authorized for a short dosing interval of 3 weeks, as this was the only dosing schedule examined in the vaccine approval trials [1, 2]. However, due to vaccine scarcity, some countries had elected to delay the second dose, so that more of the population can be covered with one dose. The strategy was further supported by prior experiences and data with other vaccines, which had shown that longer dosing intervals generally provided better vaccine immunogenicity [3–6]. Now, the impact of dosing interval on vaccine immunogenicity, durability and cross protection against variants can be carefully examined. Addressing these key questions, will inform future policies for these novel mRNA vaccines especially during times when supplies may be limited.

The Timing of Second Dose of SARS-CoV-2 mRNA Vaccine (BNT162b2) and Immunologic and FuNctional Antibody Responses Generated in Healthcare Workers (TIMING) study was a prospective observational study in Canadian healthcare workers (HCW), who were affected by the policy changes to extend the primary vaccination dosing schedule. Between January 1 to April 30, 2021, the study enrolled 3 groups of age and sex matched HCW, who received one of three dosing schedules (<35-days, 35-42-days or >42-days). The study compared the 3 dosing intervals on humoral immunity, durability and neutralizing capacity against the ancestral and immune-evasive Beta variants of SARS-CoV-2 up to 9 months post immunization.

## Methods

### Study design and participants

Consecutive HCW, who received 2 primary doses of the BNT162b2 vaccine at Hamilton Health Sciences, Ontario, Canada, between January 1 to April 30, 2021 were enrolled. All

HCW were involved in front-facing patient care, and prioritized as moderate to high risk for SARS-CoV-2 exposure by local provincial guidelines [7]. During study recruitment, the dosing interval was changed from <35-days to 35-42-days and then >42-days by the provincial government to address the scarcity in vaccine supplies. This created variations in the dosing interval that were unbiased, since HCW were unable to choose their dosing interval nor were any individual characteristics taken into consideration when implementing the different dosing schedules. All participants provided written informed consent. The study was approved by the Hamilton Research Ethics Board and conducted in accordance to the Declaration of Helsinki.

Study recruitment began during the 35-42-days dosing interval rollout, and HCW were enrolled at the time of the second dose. Concurrently, the study recruited sex- and age- (within 5 years) matched HCW into the <35 days interval group. When the provincial policy changed the dosing interval to >42 days, a third cohort of age- and sex matched HCW were enrolled (Fig 1). All participants provided blood samples at 3 weeks, 3 months and 6–9 months post-second dose. Baseline demographic and clinical information from participants, including any confirmed SARS-CoV-2 infections were recorded at all visits. Blood was collected for serum and peripheral blood mononuclear cells (PBMC) which were separated, processed and cryo-preserved at -80˚C until ready for analysis.

## Immunological assays

Binding antibody subtypes (IgM, IgG, IgA) specific to ancestral SARS-CoV-2 full-length Spike protein and Receptor Binding Domain (RBD) were measured by a validated enzyme-linked immunosorbent assay (ELISA) [8, 9]. Antibody titers were quantified by luminescence using the BioTek 800TS microplate reader; and expressed in optical density (OD). The threshold of detection for each antibody subtype has been previously established, as the mean plus 3 standard deviations above pre-COVID-19 control populations from similar geographic regions [8]. Titers above the threshold of detection were regarded as seropositive. Antibody neutralizing capacity against live viruses was assessed by a microneutralization (MNT) assay [8, 10],

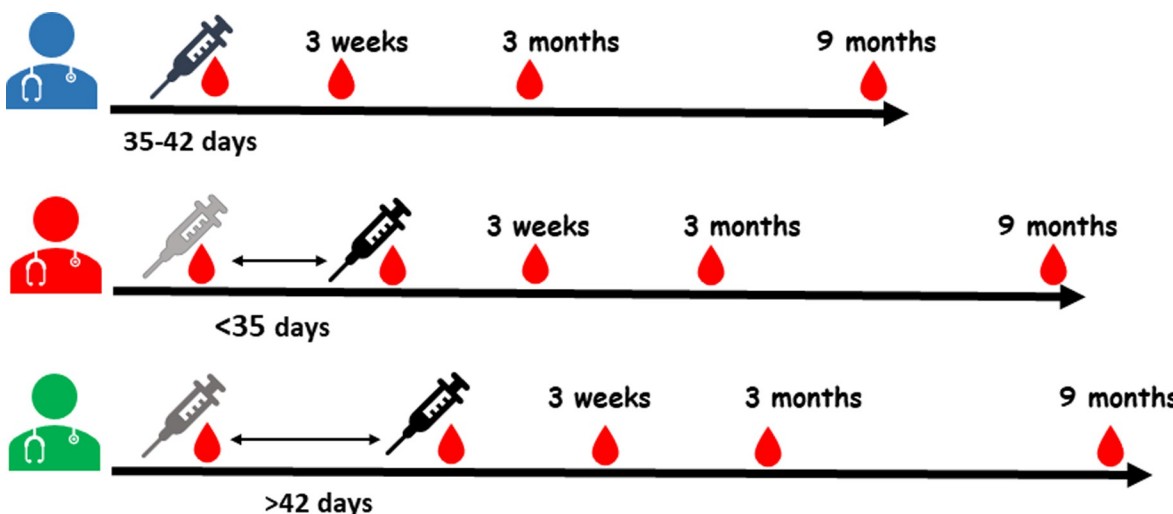

**Fig 1. Study design.** Three groups of healthcare workers (HCW) were recruited according to the timing of the second vaccine dose (black needle) at either <35 days, 35–42 days or >42 days following the first dose. Blood samples for serological tests were collected at the same timepoints from the second dose: at the time of the (pre-) second dose and at 3 weeks, 3 months and 6–9 months post-second dose. Baseline pre-first dose blood samples were available for the <35 days and >42 days groups who were recruited before the first dose, while the 35–42 days group was enrolled at the time of the second dose.

and reported in titers needed to inhibit 50% infection. SARS-CoV-2 RBD specific memory B cells were quantified from PBMC collected at 3 months post-second dose, using a commercially available ELISPOT kit (Mabtech). The frequency of specific memory B cells were calculated from the ratio of cells secreting SARS-CoV-2 RBD specific IgG versus all IgG-secreting cells. For details of all the immunological assays and their threshold of detection, please refer to S1 Appendix in S1 File.

### Statistical analysis

All participants reporting COVID-19 infection confirmed by locally preferred microbiological methods, prior to enrolment were excluded. Participants who developed (confirmed) COVID-19 during follow-up were censored at the time of the infection (S2 Appendix in S1 File). There were no missing data. Normally distributed variables were summarized as means and standard deviation (SD) and non-normally distributed variables as median and $25^{th}$-$75^{th}$ percentile. Baseline characteristics were compared using analysis of variance (for normally distributed variables), Kruskal-Wallis (non-normally distributed variables) or Fisher's exact (categorical variables) tests. Neutralizing antibody titers were log-transformed prior to analysis. Comparison of titers across timepoints between groups were performed with linear mixed effects model adjusted for age, sex, education, smoking, alcohol use, body-mass index, and comorbidities (coronary artery disease, strokes, heart failure, atrial fibrillation, previous cancer, autoimmune disease, renal disease, chronic obstructive pulmonary disease and asthma). To assess for differences in antibody kinetics between groups, an interaction term, groups*visits was included in the model. Where the interaction term was significant at $p < 0.05$, post hoc comparison to evaluate for differences in titers between groups at each time point was conducted. Details for sample size calculations are provided in S3 Appendix in S1 File. All analyses were conducted using STATA 17 (StataCorp LLC, Texas, USA).

## Results

The study enrolled 313 HCW, of whom, 4 (1.3%) were excluded due to prior self-reported baseline COVID-19 infection. Baseline characteristics of the remaining 309 participants were similar between groups (Table 1). In all groups, there were more females, non-smokers, and middle-aged adults. Many of the participants were highly educated and reported low burden of comorbidities. For all immunological outcomes, the interaction term between groups*visits were highly significant ($p < 0.0001$); suggesting significant variation in antibody kinetics between groups.

### Ancestral SARS-CoV-2 anti-Spike and anti-RBD IgG titers

Mean anti-Spike and anti-RBD IgG titers and IgG seropositive rates by visits and groups are shown in Fig 2 (adjusted values) and S4 Appendix in S1 File (crude values). Prior to the second dose, mean anti-Spike and anti-RBD IgG titers were lowest in the >42-days group, suggesting significant waning of IgG levels beyond 42 days following the first dose. The seropositive rates for anti-SARS-CoV-2 IgG were over 90% in each group post-first dose, and remained high for anti-Spike IgG beyond 42 days; while anti-RBD IgG fell to 65% after 42 days. The second dose induced a significant and consistent (100% seroconversion in all groups) rise in all IgG titers, at 3 weeks, which then declined to pre-second dose titers by 6–9 months. Between-group comparison showed no significant difference in anti-Spike IgG titers at 3 weeks or 3 months. However, by 6–9 months, the >42-days group had higher mean titer compared to the <35-days group (2.14 [95%CI 1.86–2.62] *vs.*1.89 [1.50–2.32], $p < 0.0001$). By contrast, the >42-days group showed significantly higher anti-RBD IgG titers, compared to the <35-days group at 3

**Table 1. Baseline characteristics of participants by dosing interval.**

| Characteristics | | Time interval between 1st and 2nd vaccine doses | | | |
|---|---|---|---|---|---|
| | | <35 days (n = 108) | 35–42 days (n = 102) | >42 days (n = 99) | p-value |
| Median days between vaccine doses* | | 21 (21–21) | 39 (39–39) | 83 (73–98) | - |
| Age, years ± SD | | 43±13 | 40±13 | 42±13 | 0.18 |
| Body-mass index, kg/m2 ± SD | | 28.7±5.9 | 27.3±5.8 | 26.9±5.7 | 0.062 |
| Sex | Female | 63 (58%) | 55 (54%) | 57 (58%) | 0.86 |
| | Male | 45(42%) | 46 (45%) | 41 (41%) | |
| | Prefer not to say | 0 (0%) | 1 (1%) | 1 (1%) | |
| Education | High school | 8 (8%) | 4 (4%) | 7 (7%) | 0.51 |
| | College | 59 (55%) | 62 (61%) | 52 (53%) | |
| | Graduate school | 40 (37%) | 36 (35%) | 40 (40%) | |
| Smoking | Never | 82 (76%) | 79 (77%) | 76 (77%) | 0.99 |
| | Current | 7 (6%) | 5 (5%) | 6 (6%) | |
| | Former | 19 (18%) | 18 (18%) | 17 (17%) | |
| Alcohol | Never | 27 (25%) | 23 (23%) | 19 (19%) | 0.75 |
| | Current | 77 (71%) | 75 (73%) | 78 (79%) | |
| | Former | 4 (4%) | 4 (4%) | 2 (2%) | |
| Asthma | | 9 (8%) | 14 (14%) | 10 (10%) | 0.46 |
| COPD | | 1 (1%) | 0 (0%) | 0 (0%) | 1.00 |
| Previous pneumonia | | 0 (0%) | 1 (1%) | 1 (1%) | 0.54 |
| Autoimmune disease | | 7 (6%) | 3 (3%) | 4 (4%) | 0.46 |
| Renal disease | | 2 (2%) | 0 (0%) | 0 (0%) | 0.33 |
| Cardiovascular disease | | 5 (5%) | 1 (1%) | 1 (1%) | 0.23 |

Data are presented as mean ± SD,

*median (25th-75th percentile) or count (column percentage). COPD = chronic obstructive pulmonary disease. P-values are for comparisons between groups, using analysis of variance (for normally distributed variables), Kruskal-Wallis (non-normally distributed variables) or Fisher's exact (categorical variables) tests. Self-reported cardiovascular disease includes physician-diagnosed angina, myocardial infarction, strokes, heart failure, or atrial fibrillation. Chronic respiratory disease includes asthma or chronic obstructive pulmonary disease.

months (2.65 [2.08–2.88] *vs*. 2.14 [1.66–2.65], p = 0.0002) and 6–9 months (0.79 [0.63–1.23] *vs*. 0.67 [0.45–0.88], p = 0.0010). The 35-42-days group had similar anti-Spike and anti-RBD IgG responses as the <35-days group post-second dose. The seropositive rates for anti-Spike IgG were >98% in all groups, up to 6–9 months; while seropositivity for anti-RBD IgG had dropped to 70%, 62% and 84% in the <35, 35–42 and >42-days groups, respectively by 6–9 months.

**Ancestral SARS-CoV-2 anti-Spike and anti-RBD IgA and IgM titers**

Vaccine induced IgA responses were similar to IgG, albeit lower in magnitude and durability (Fig 3 and S4 Appendix in S1 File). There was significant waning in IgA titers during longer intervals between doses. The lowest mean titer was observed in the >42-days group prior to the second dose. The second dose induced a significant rise in all anti-SARS-CoV-2 IgA titers across groups at 3 weeks. This was followed by a rapid decline in titers, which reached pre-second dose levels at 6–9 months anti-Spike IgA and at 3 months for anti-RBD IgA. Between-group comparisons showed no difference in mean anti-Spike IgA titers for all timepoints post-second dose. In contrast, peak anti-RBD IgA titers at 3 weeks were significantly lower in the 35-42-days and >42-days groups (compared to <35-days group), but these differences were no longer apparent after 3 months. Seropositivity for anti-Spike IgA post-second dose at 3

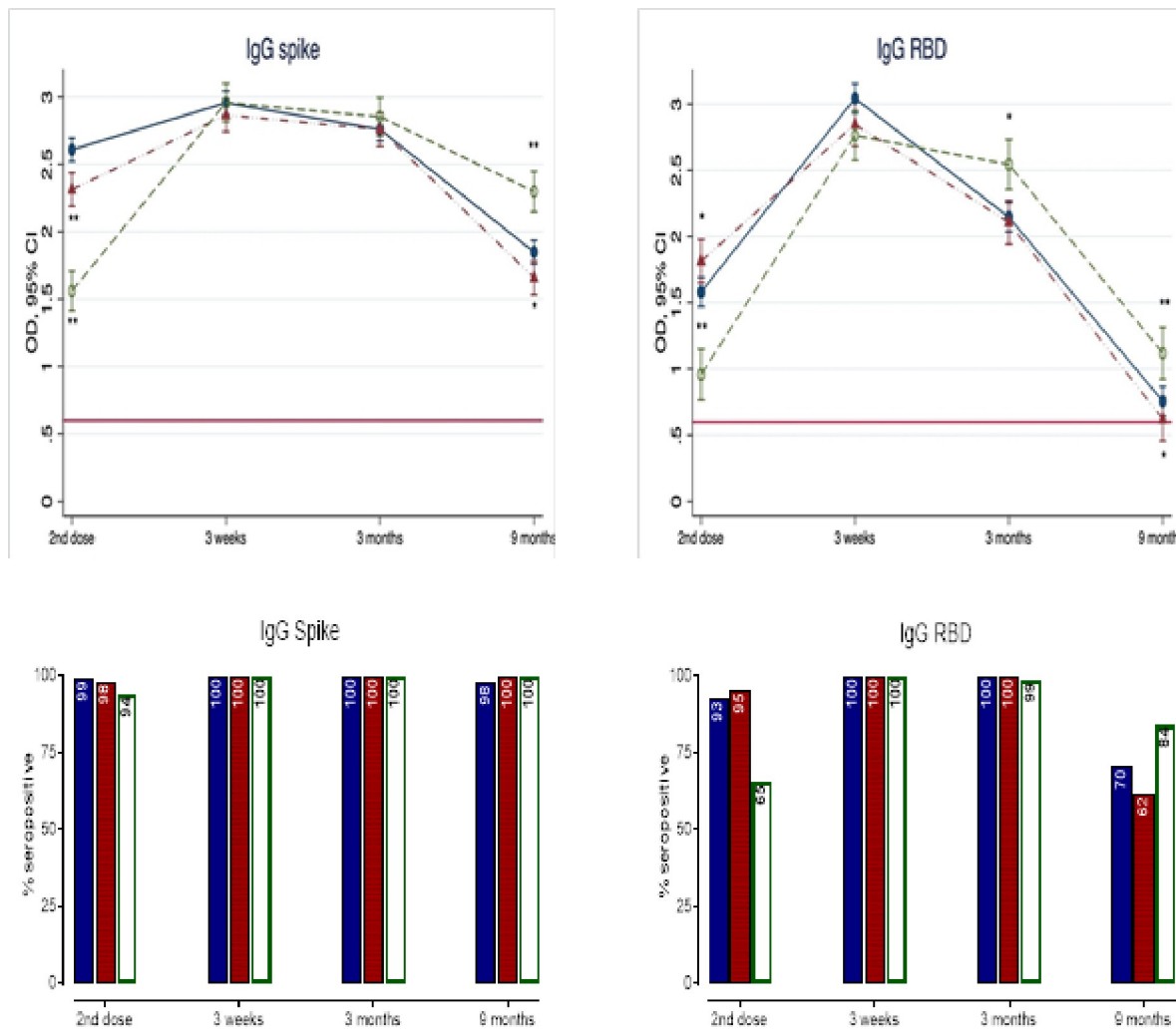

**Fig 2. Mean titers and seropositivity rates for IgG to ancestral SARS-CoV-2 Spike and RBD by study visit and dosing interval.** Upper panel: Serum IgG titers to ancestral SARS-CoV-2 Spike and RBD measured in each group at the same time point post-2nd dose using a validated ELISA (see METHODS). Data are expressed in optical density (OD) with 95% confidence interval (CI) and fully adjusted for age, sex, education, smoking, alcohol use, body-mass index, known comorbidities (cardiovascular, neoplastic, autoimmune, renal and chronic respiratory diseases) and the number of days after 2nd vaccine dose to subsequent blood draw. The dosing interval groups 35-42-days (red symbols) and >42-days (open green symbols), were compared to the <35-days group (reference, blue closed symbols). For the summary numerical data, please refer to S3 Appendix in S1 File. Interaction terms for visit x group were all significant at p<0.0001. Therefore, between group comparisons (<35-day group as reference) were conducted with p-values *<0.05 and **<0.005. The red solid line represents the assay's threshold of detection (see S1Appendix in S1 File for numerical values). Lower Panel: The bars represent the % of participants with seropositive IgG to ancestral SARS-CoV-2 Spike and RBD within each group by visit. Seropositivity was defined by titer levels above the assay's threshold of detection.

weeks was >97% in all groups, and dropped similarly across groups to 83–94% at 3 months and to 41–45% by at 6–9 months. The seropositive rates for anti-RBD IgA were overall lower (than anti-Spike), and rapidly declined to only 2–6% at 6–9 months across groups.

Lower vaccine induced anti-Spike and anti-RBD IgM mean titers and seropositive rates were observed in all groups, following the first and second dose (Fig 4 and S4 Appendix in S1 File). Specifically, IgM titers waned quickly following the first dose, with the lowest mean titer observed in the >42-days group prior to the second dose. The main difference between groups was observed at 3 weeks, with the highest IgM titer demonstrated in the <35-days group.

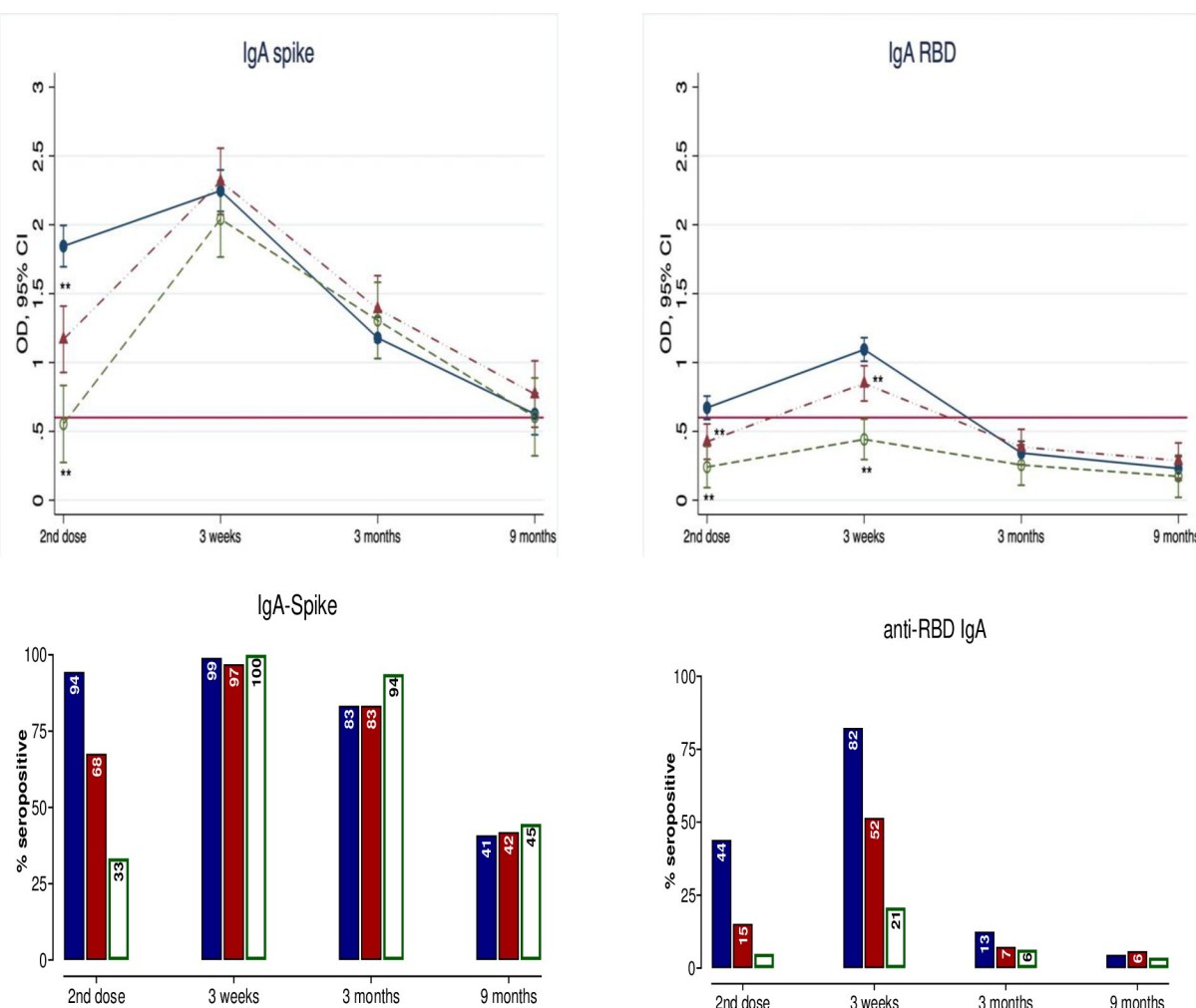

**Fig 3. Mean titers and seropositivity rates for IgA to ancestral SARS-CoV-2 Spike and RBD by study visit and dosing interval.** Upper panel: Serum IgA titers to ancestral SARS-CoV-2 Spike and RBD measured in each group at the same time points post-2nd dose using a validated ELISA (see METHODS). Data are expressed in optical density (OD) with 95% confidence interval (CI) and fully adjusted for age, sex, education, smoking, alcohol use, body-mass index, known comorbidities (cardiovascular, neoplastic, autoimmune, renal and chronic respiratory diseases) and the number of days after 2nd vaccine dose to subsequent blood draw. The dosing interval groups 35-42-days (red symbols) and >42-days (open green symbols), were compared to the <35-days group (reference group, blue closed symbols). For the summary numerical data, please refer to S3 Appendix in S1 File. Interaction terms for visit x group were all significant at p<0.0001. Therefore, between group comparisons (<35-day group as reference) were conducted with p-values *<0.05 and **<0.005. The red solid line represents the assay's threshold of detection (see S1 Appendix in S1 File for numerical values). Lower Panel: The bars represent the % of participants with seropositive IgA to ancestral SARS-CoV-2 Spike and RBD within each group by visit. Seropositivity was defined by titer levels above the assay's threshold of detection.

Seropositivity for anti-Spike and anti-RBD IgM were low following the second dose and declined markedly over 3 and 6–9 months particularly for anti-RBD IgM.

## Neutralizing titers against SARS-CoV-2 ancestral and beta variant

Neutralizing titers against ancestral SARS-CoV-2 prior to the second dose were not significantly different between the 35-42-days (log mean 3.69 [3.00–4.38]) and <35-days groups (4.38 [3.00–5.08]), both of which were significantly higher than the >42-days group (2.30 [1.61–3.00]) (Fig 5 and S5 Appendix in S1 File). The second dose induced a significant rise in neutralizing titers in all groups, which then declined to levels comparable to pre-second dose

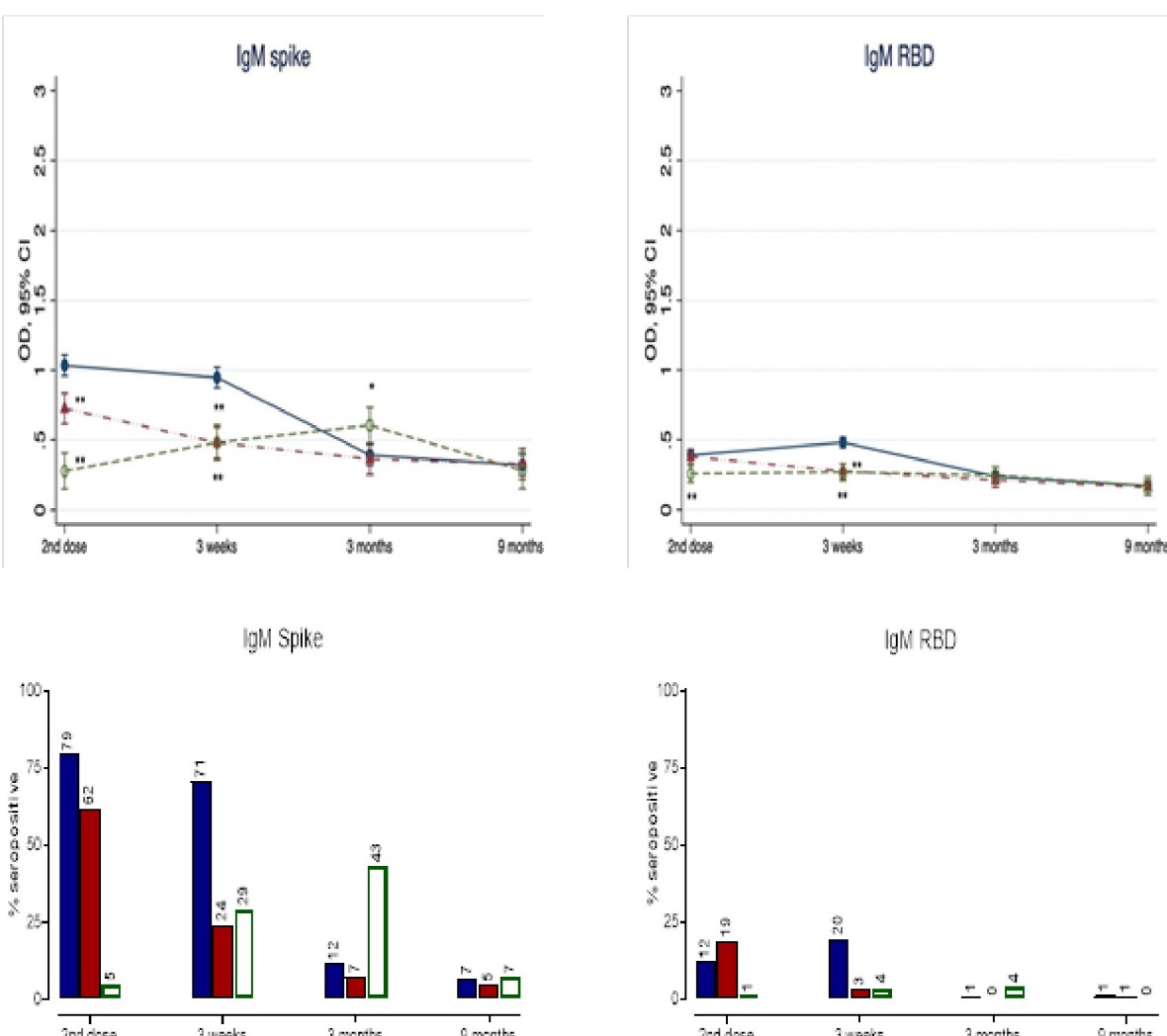

**Fig 4. Mean titers and seropositivity rates for IgM to ancestral SARS-CoV-2 Spike and RBD by study visits and dosing interval groups.**
Upper panel: Serum IgM titers to ancestral SARS-CoV-2 Spike and RBD measured in each group at the same time points post-2nd dose using a validated ELISA (see METHODS). Data are expressed in optical density (OD) with 95% confidence interval (CI) and fully adjusted for age, sex, education, smoking, alcohol use, body-mass index, known comorbidities (cardiovascular, neoplastic, autoimmune, renal and chronic respiratory diseases) and the number of days after 2nd vaccine dose to subsequent blood draw. The dosing interval groups were 35-42-days (red symbols) and >42-days (open green symbols), who were compared to the <35-days group (reference group, blue closed symbols). For the summary numerical data, please refer to S3 Appendix in S1 File. Interaction terms for titer levels x visit x group were all significant at p<0.0001. Therefore, between group comparisons (<35-day group as reference) were conducted with p-values *<0.05 and **<0.005. The red solid line represents the assay's threshold of detection (see S1 Appendix in S1 File for numerical values). Lower Panel: The bars represent the % of participants with seropositive IgM to ancestral SARS-CoV-2 Spike and RBD within each group by visit. Seropositivity was defined by titer levels above the assay's threshold of detection.

titers by 3 months. For all timepoints post-second dose, neutralizing titers against ancestral SARS-CoV-2 were highest in the >42-days group. For example, neutralizing titers against the ancestral strain in the <35-days versus >42-days groups were log mean 4.38 [3.69–5.08] *vs.* 5.08 [4.38–5.77], p<0.0001 at 3 months; and 2.30 [1.61–3.00] *vs.* 3.00 [2.30–3.69], p = 0.0007 at 6–9 months.

A similar pattern of response was observed for neutralizing titers against the immune evasive SARS-CoV-2 Beta (B.1.351) variant. There was significant waning in neutralizing titers post first dose, and the >42-days group had the lowest neutralizing capacity against

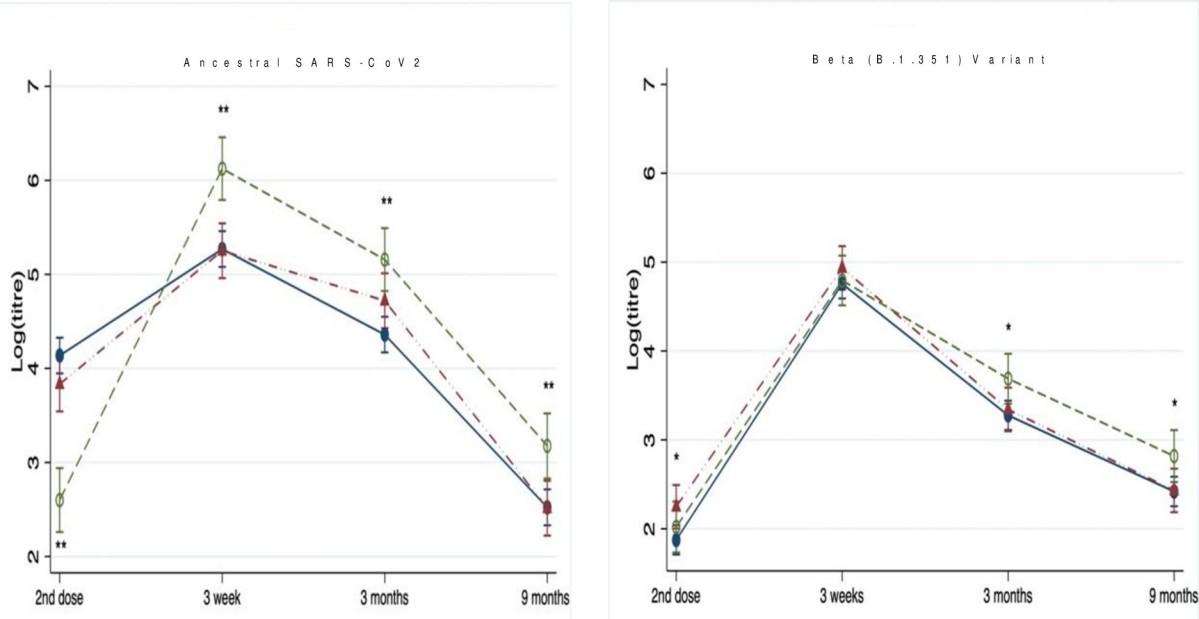

**Fig 5. Neutralizing titers to live ancestral SARS-CoV-2 and Beta variant by visits stratified by dosing interval groups.** Data are presented as logarithmic titers (means and 95% CI) needed to inhibit 50% of infection due to Ancestral or Beta variants after fully adjusted for age, sex, education, smoking, alcohol use, body-mass index, known comorbidities (cardiovascular, neoplastic, autoimmune, renal and chronic respiratory diseases) and the number of days after 2nd vaccine dose to subsequent blood draw. The dosing interval groups were 35-42-days (red symbols) and >42-days (open green symbols), who were compared to the <35-days group (reference group, blue closed symbols). For the summary numerical data, please refer to S4 Appendix in S1 File. Interaction terms for titer levels x visit x group was significant at p<0.0001. Therefore, between group comparisons (<35-day group as reference) were conducted with p-values *<0.05 and **<0.005.

SARS-CoV-2 B.1.351 variant prior to the second dose. At 3 weeks post-second dose, there was no difference in peak titers between groups. However, beyond this timepoint at 3 and 6–9 months, the >42-days group had significantly higher neutralizing capacity against SARS-CoV-2 Beta variant.

### Anti-RBD IgG secreting memory B cells

In a small random subset of 30 participants, PBMC collected at 3 months post-second dose were quantified for anti-RBD IgG specific memory B cells. Due to the small sample size, the <35-days and 35-42-days groups were combined and compared to the >42-days group. After adjusting for potential confounders, between group comparison showed significantly higher mean RBD-specific memory B cells in the >42-days group versus the shorter interval group (0.76 [SD 0.68] *vs.* 0.305 [SD 0.282], p = 0.016) (Fig 6).

### Discussion

In a healthy cohort of HCW, 3 dosing intervals for the primary vaccination of the BNT162b2 mRNA COVID-19 vaccine on the humoral response were compared. We observed a significant effect of extending the dosing interval beyond 42 days. We showed that (1) binding and neutralizing antibody titers waned dramatically after 35 days following the first dose; (2) the second dose induced a robust antibody response with all dosing schedules, which peaked at 3 weeks and declined thereafter to pre-second dose levels by 6–9 months; (3) secondary IgG response against ancestral SARS-CoV-2 full length Spike was more robust and durable than to RBD; and IgG response was greater in magnitude and durability than IgA and IgM responses;

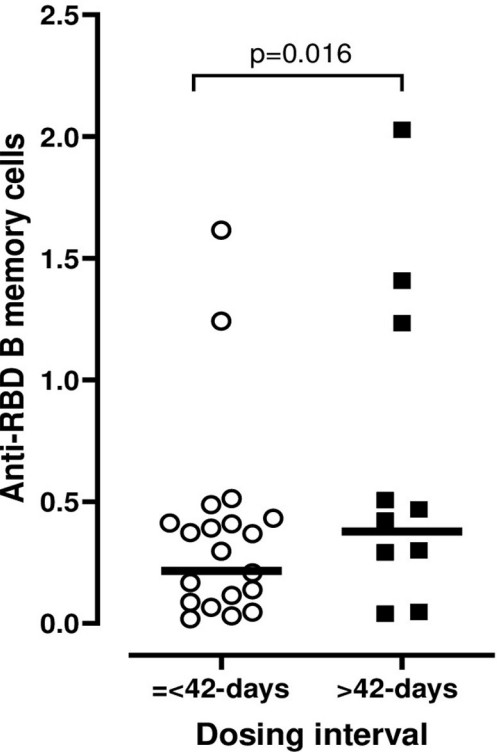

**Fig 6. Anti-RBD memory B cells at 3 months according to dosing interval = <42 days or >42 days between doses.**
In 30 random participants anti-RBD IgG specific B memory cells were measured using commercially available
ELISPOT kit (Mabtech). Data are plotted for participants with dosing interval = <42 days (<35-days and 35-42-days
groups combined) and >42-days group with the adjusted mean provided for each group. Comparison was performed
using linear mixed effects model adjusted for age, sex, education, smoking, alcohol use, body-mass index, known
comorbidities (cardiovascular, neoplastic, autoimmune, renal and chronic respiratory diseases) and the number of
days after 2nd vaccine dose to subsequent blood draw.

(4) compared to shorter intervals, a long dosing interval of >42-days provided higher and
more sustained binding and neutralizing antibody titers against SARS-CoV-2 ancestral and
Beta variants beyond 3 months after immunization.

In animal models, extending the dosing interval beyond 4 weeks has been shown to potenti-
ate B cell responses, by selectively allowing clonal expansion and differentiation of high affinity
B cells into antibody-producing plasma cells and memory B cells [3, 11, 12]. This aligns with
clinical evidence supporting higher vaccine efficacy with longer dosing intervals usually
beyond 6 weeks [6]. Moreover, dosing intervals of less than 3 weeks may impair the develop-
ment of a robust immune response, as high-levels of pre-existing antibodies (from the previous
dose) have been shown to negatively correlate with antibody responses to vaccination [13–15].
Real world data from the UK and Canada have now emerged supporting the population-level
benefit of extending the dosing interval to 12 weeks for the primary vaccination series of
COVID-19 [16–18]. In both young and older adults, delaying the second dose to 12 weeks was
associated with significantly lower transmission and symptomatic infections from ancestral
SARS-CoV-2. This was further corroborated by data in a small sample of participants, who
had higher anti-Spike and anti-RBD IgG titers with dosing intervals >42 days compared to
<30 days at 3 months post immunization. Our data adds to this evidence and extends the find-
ings to include an intermediate dosing interval of 35-42-days which has not been well studied.
Furthermore, we provided new data on antibody kinetics up to 9 months following the second

dose to inform the impact of dosing interval on the durability of vaccine-induced antibody titers. Similar to previous reports, we found that the standard interval of <35-days provided a robust antibody response following the second dose, which reached a peak at 3 weeks; and began to decline shortly thereafter to lower levels by 3 months [17, 19, 20]. We found no meaningful difference between the <35-days and 35-42-days groups in antibody responses following the second dose. In contrast, the >42-days group showed a much slower rate of decline in antibody titers, and therefore, provided more sustained higher antibody levels beyond 3 months. Furthermore, significantly higher level of anti-RBD specific B memory cells were generated at 3 months in this group, suggesting the development of a robust humoral memory response.

Current evidence suggests that protection against ancestral SARS-CoV-2 infections following the first dose may primarily be mediated by non-neutralizing antibodies (Fc-dependent effector functions) and virus specific T cells [21]; while heterotypic protection against infection with SARS-CoV-2 variants may require neutralizing antibodies [22]. This is becoming more important as more immune evasive variants such as Omicron emerge. Our findings indicate that there was a significant effect of the longer dosing interval on the durability of neutralizing antibody titers against immune evasive Beta variants. This suggests that longer dosing intervals may provide more durable immune protection against emerging variants and help extend the interval between subsequent booster doses.

In this study, we also examined vaccine induced IgA and IgM responses to BNT162b2 vaccination, which have not been well documented previously. Due to its localization at mucosal surfaces, it has been proposed that IgA may prevent acquisition and transmission of infection [23]. In the context of mRNA COVID-19 vaccines, our findings align with one other study [24], which showed that the vaccine induced IgA response, while qualitatively similar to IgG, is smaller in magnitude and durability. Lastly, we found that most participants did not produce a robust IgM response to vaccination at the timepoints examined and that IgM levels waned substantially. However, IgM has been reported to play a significant role in neutralization mediated by convalescent plasma [25].

Our findings have a number of potential implications. First, there may be a benefit of generating a more sustained higher binding and neutralizing titer against ancestral SARS-CoV-2 beyond 3 months after vaccination with a longer dosing interval of >42 days. The trade-off, however, is the potentially higher risk of infection following the first dose, particularly to highly transmissible variants as binding and neutralizing titers wane with longer intervals between doses. Second, the differences in neutralizing titers against immune-evasive beta variant between study groups suggest that the primary vaccination dosing interval may have an impact on infections with immune-evasive variants. Nonetheless, the overall waning in neutralizing titers against both ancestral and immune-evasive beta variant beyond 3 months suggest that immunity to highly transmissible and antigenically distinct variants (like omicron) may not be sufficiently high or durable with the current primary vaccination schedule comprising of two BNT162b2 vaccine doses to mediate effective protection.

The strengths of our study include the large sample size, the well-balanced groups of sex and age matched participants with similar demographic and risk levels, and the long follow-up with comparable timepoints of comparison across groups. Furthermore, the allocation of HCW to the dosing schedule was unbiased and we were able to adjust for many potential confounders. All of these factors help to minimize potential biases and allows for valid comparison and unbiased estimation of the immune effects between different dosing intervals. The limitations include the predominantly young and healthy white population with no prior SARS-CoV-2 infections, which limits the generalizability of our findings to other populations with different demographics. This would also include a proportion of the population with hybrid

immunity from prior or intercurrent SARS-CoV-2 infections, who are represented in this study. All of our participants received the BNT162b2 vaccine and therefore our findings may not be generalizable to other COVID-19 vaccines. Although vaccine induced antibody responses may be correlative to immune protection, our study was not powered to measure vaccine effectiveness. Furthermore, our binding antibody ELISA had an upper limit of detection, which may limit our ability to detect differences between groups, when titers exceed the upper limit. Lastly, the immune evasive Beta variant (B.1.351) assessed in our study was not one of the more dominant strain, but nonetheless is one of the more difficult variants to neutralize. Therefore, the implications of our data on the durability of neutralizing titers against emerging immune evasive variants remains important.

In conclusion, we found that extending the primary vaccination dosing schedule beyond 42 days had a significant impact on inducing higher SARS-CoV2 anti-Spike and RBD antibody titers and neutralizing capacity up to 6–9 months post vaccination to both SARS-CoV-2 ancestral and Beta variant. This information could help to inform decisions regarding optimal dosing interval that will address local viral epidemiology while balancing population needs and vaccine availability.

## Supporting information

**S1 File. Contains all the supporting tables and figures.**
(DOCX)

## Acknowledgments

We would like to acknowledge the contributions of Lucas Bilaver, Braeden Cowborough, Delia Heroux, Amy Moorehead, Andrea Hucik, Breanna Landry, Nick Ivetic, Lesly Wiltshire, Jennifer Wattie, Vanessa Sabourin, Samantha Marino and Alexander Georgiou for the collection, processing and analysis of blood samples; and to Roxanna Solanno, Sandy Trottier, Bushra Bhatti, Ali Moinuddin and Tessa Anzai for their assistance in participant recruitment and study visits. Our sincerest appreciation to our research participants, for their time and commitment to the study.

## Author Contributions

**Conceptualization:** Darryl P. Leong, Mark Loeb, Judah A. Denburg, Dominik Mertz, Zain Chagla, Ishac Nazy, Matthew S. Miller, Dawn M. E. Bowdish, MyLinh Duong.

**Data curation:** Darryl P. Leong, Ali Zhang, Jessica A. Breznik, Maha Mushtaha, Sumathy Rangarajan, Hannah Stacey, MyLinh Duong.

**Formal analysis:** Darryl P. Leong, Ali Zhang, MyLinh Duong.

**Funding acquisition:** MyLinh Duong.

**Investigation:** Ali Zhang, Jessica A. Breznik, Rumi Clare, Angela Huynh, Paul Y. Kim, Ishac Nazy, Matthew S. Miller, Dawn M. E. Bowdish.

**Methodology:** Ali Zhang, Jessica A. Breznik, Rumi Clare, Angela Huynh, Hannah Stacey, Paul Y. Kim, Ishac Nazy, Matthew S. Miller, Dawn M. E. Bowdish, MyLinh Duong.

**Project administration:** Rumi Clare, Angela Huynh, Maha Mushtaha, Sumathy Rangarajan, Hannah Stacey, Paul Y. Kim, Dominik Mertz, MyLinh Duong.

**Resources:** Sumathy Rangarajan.

**Supervision:** Jessica A. Breznik, Maha Mushtaha, Mark Loeb, Judah A. Denburg, Dominik Mertz, Zain Chagla, Ishac Nazy, Matthew S. Miller, Dawn M. E. Bowdish, MyLinh Duong.

**Validation:** Darryl P. Leong, Maha Mushtaha, Sumathy Rangarajan, Paul Y. Kim, Mark Loeb, Ishac Nazy, Dawn M. E. Bowdish, MyLinh Duong.

**Visualization:** Dawn M. E. Bowdish, MyLinh Duong.

**Writing – original draft:** Darryl P. Leong, MyLinh Duong.

**Writing – review & editing:** Darryl P. Leong, Ali Zhang, Jessica A. Breznik, Rumi Clare, Angela Huynh, Maha Mushtaha, Sumathy Rangarajan, Hannah Stacey, Paul Y. Kim, Mark Loeb, Judah A. Denburg, Dominik Mertz, Zain Chagla, Ishac Nazy, Matthew S. Miller, Dawn M. E. Bowdish, MyLinh Duong.

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
