## [Decision Letter · Decision Letter 0]

18 Nov 2022

PONE-D-22-28685Comparison of three dosing intervals for the primary vaccination of the SARS-CoV-2 mRNA Vaccine (BNT162b2) on magnitude, neutralization capacity and durability of the humoral immune response in health care workers: a prospective cohort study.PLOS ONE

Dear Dr. Duong,

Thank you for submitting your manuscript to PLOS ONE. After careful consideration, we feel that it has merit but does not fully meet PLOS ONE’s publication criteria as it currently stands. Therefore, we invite you to submit a revised version of the manuscript that addresses the points raised during the review process.

We look forward to receiving your revised manuscript.

Kind regards,

Glenda Canderan, PhD

Academic Editor

PLOS ONE

Additional Editor Comments:

Minor comment: I believe the image in figure 1 represents a veterinary and not a health care worker. Please modify for clarity

Reviewers' comments:

Reviewer's Responses to Questions

**Comments to the Author**

1. Is the manuscript technically sound, and do the data support the conclusions?

Reviewer #1: Yes

Reviewer #2: Yes

2. Has the statistical analysis been performed appropriately and rigorously? 

Reviewer #1: Yes

Reviewer #2: Yes

3. Have the authors made all data underlying the findings in their manuscript fully available?

Reviewer #1: No

Reviewer #2: No

4. Is the manuscript presented in an intelligible fashion and written in standard English?

Reviewer #1: Yes

Reviewer #2: Yes

5. Review Comments to the Author

Reviewer #1: This is an interesting study that compared Ab responses to BNT162b2 according to the dosing schedule for the 1st and 2nd shot. The question of timing is relevant, especially for ongoing development and optimization of mRNA vaccination strategies.

My major comment is that the ELISA read-out is non-standardized, and it is hard to extrapolate these findings (reported as OD units) to studies that used more traditional assays with read-outs in BAU/mL, micrograms/mL, etc. I also have some question with the statements about differences in levels of Ab to spike vs RBD. Given that the antigen in the vaccine is the S-RBD and not full-length spike, it seems doubtful that the Ab are actually higher to spike than RBD. It should at least be acknowledged that this could be an artifact related to the details of the assays (eg, quantity and quality of the proteins used on the sold phase).

There are a few minor language and grammar aspects that could be improved.

Finally, it is not clear that the PLoS Data policy is being adhered to.

Reviewer #2: Duong and colleagues have reported on a prospective observational study comparing different intervals between the first and second doses of the Pfizer/BioNTech mRNA vaccine. The study enrolled healthcare workers who received their second dose at different intervals in accordance with shifting guidance from provincial health authorities. While not randomized, the participants could not choose their interval, and the groups were well-matched overall. Peak binding antibody titres were not different amongst the groups, perhaps because of a ceiling artefact in their ELISA assay. Neutralizing activity was higher at the peak time point in the longest interval group. Durability of immune responses was consistently improved in the longest interval group, including against the Beta variant. Their data adds rigor to the population-level observations that longer intervals in the primary vaccine series leads to higher (and broader) antibody titres against SARS-CoV-2.

Major Criticisms: None.

Minor Criticisms:

How was the PBMC subset identified? Were PBMC collected from all individuals and then a randomization procedure was done to determine which participants’ samples should be thawed and assayed?

While the Beta variant never spread globally, it is a representative of difficult to neutralize variants. The manuscript would be more timely if the authors had looked at more recent variants, but this could be mentioned as a limitation in their Discussion.

The widespread deployment of boosters and the high prevalence of prior or intercurrent infection means that most people in developed countries have “hybrid immunity” which further limits the generalizability of their work. This too should be listed as a limitation.

6. PLOS authors have the option to publish the peer review history of their article (what does this mean?). If published, this will include your full peer review and any attached files.

Reviewer #1: No

Reviewer #2: **Yes: **Stephen Walsh

---

## [Author Response · Author response to Decision Letter 0]

27 Dec 2022

PONE-D-22-28685

Comparison of three dosing intervals for the primary vaccination of the SARS-CoV-2 mRNA Vaccine (BNT162b2) on magnitude, neutralization capacity and durability of the humoral immune response in health care workers: a prospective cohort study.

We thank the reviewers for their time and insightful comments, which we have addressed below. Where appropriate the suggested changes and comments have been added to the revised manuscript, which have strengthened this paper. 

We hope that our responses and changes to the manuscript address all the reviewers’ and editor’s concerns. We thank you in advance for taking the time to consider our work.

Reviewer #1: This is an interesting study that compared Ab responses to BNT162b2 according to the dosing schedule for the 1st and 2nd shot. The question of timing is relevant, especially for ongoing development and optimization of mRNA vaccination strategies.

>> We thank the reviewer for these positive comments. 

My major comment is that the ELISA read-out is non-standardized, and it is hard to extrapolate these findings (reported as OD units) to studies that used more traditional assays with read-outs in BAU/mL, micrograms/mL, etc. 

>> We acknowledge the value of reporting in BAU/ml; however the ELISA assay utilized in this study was developed by our group and has used extensively (PMID: 34504336, 36426948, 36137590, 34835045, 34736891). We feel that it is appropriate to use this metric to make comparisons between groups as it is internally valid within this study and also consistent with our other work in this area. 

I also have some question with the statements about differences in levels of Ab to spike vs RBD. Given that the antigen in the vaccine is the S-RBD and not full-length spike, it seems doubtful that the Ab are actually higher to spike than RBD. It should at least be acknowledged that this could be an artifact related to the details of the assays (eg, quantity and quality of the proteins used on the sold phase).

>>To our knowledge, the Comirnaty/Pfizer BioTech vaccine includes mRNA encoding the sequence of the pre-fusion stabilized spike protein (see https://www.nature.com/articles/s41541-021-00369-6 and the product monograph). The RBD domain is a small fraction of the complete spike protein. Therefore vast majority of antibodies made in response to the vaccine are therefore directed against non-RBD domains.

There are a few minor language and grammar aspects that could be improved. 

>> We have gone through the paper and made some changes to the language and grammar aspects of the paper. We hope that this has improved the readability of the paper. 

Finally, it is not clear that the PLoS Data policy is being adhered to.

>> the data access statement has been revised to comply with PLoS Data policy.

Reviewer #2: Duong and colleagues have reported on a prospective observational study comparing different intervals between the first and second doses of the Pfizer/BioNTech mRNA vaccine. The study enrolled healthcare workers who received their second dose at different intervals in accordance with shifting guidance from provincial health authorities. While not randomized, the participants could not choose their interval, and the groups were well-matched overall. Peak binding antibody titres were not different amongst the groups, perhaps because of a ceiling artefact in their ELISA assay. Neutralizing activity was higher at the peak time point in the longest interval group. Durability of immune responses was consistently improved in the longest interval group, including against the Beta variant. Their data adds rigor to the population-level observations that longer intervals in the primary vaccine series leads to higher (and broader) antibody titres against SARS-CoV-2.

Major Criticisms: None.

Minor Criticisms:

How was the PBMC subset identified? Were PBMC collected from all individuals and then a randomization procedure was done to determine which participants’ samples should be thawed and assayed?

>> All participants had blood collected for serology and PBMC, which were stored. We then randomly select 30 specimens to thaw and analyzed.

While the Beta variant never spread globally, it is a representative of difficult to neutralize variants. The manuscript would be more timely if the authors had looked at more recent variants, but this could be mentioned as a limitation in their Discussion.

>> we agree and this has been added as one of the limitations in the discussion. 

The widespread deployment of boosters and the high prevalence of prior or intercurrent infection means that most people in developed countries have “hybrid immunity” which further limits the generalizability of their work. This too should be listed as a limitation.

>> we agree and this has been added as one of the limitations in the discussion.

---

## [Decision Letter · Decision Letter 1]

30 Jan 2023

Comparison of three dosing intervals for the primary vaccination of the SARS-CoV-2 mRNA Vaccine (BNT162b2) on magnitude, neutralization capacity and durability of the humoral immune response in health care workers: a prospective cohort study.

PONE-D-22-28685R1

Dear Dr. Duong,

We’re pleased to inform you that your manuscript has been judged scientifically suitable for publication and will be formally accepted for publication once it meets all outstanding technical requirements.

Kind regards,

Glenda Canderan, PhD

Academic Editor

PLOS ONE

Additional Editor Comments (optional):

Reviewers' comments:

Reviewer's Responses to Questions

**Comments to the Author**

1. If the authors have adequately addressed your comments raised in a previous round of review and you feel that this manuscript is now acceptable for publication, you may indicate that here to bypass the “Comments to the Author” section, enter your conflict of interest statement in the “Confidential to Editor” section, and submit your "Accept" recommendation.

Reviewer #1: All comments have been addressed

Reviewer #2: All comments have been addressed

2. Is the manuscript technically sound, and do the data support the conclusions?

Reviewer #1: (No Response)

Reviewer #2: Yes

3. Has the statistical analysis been performed appropriately and rigorously? 

Reviewer #1: (No Response)

Reviewer #2: Yes

4. Have the authors made all data underlying the findings in their manuscript fully available?

Reviewer #1: (No Response)

Reviewer #2: Yes

5. Is the manuscript presented in an intelligible fashion and written in standard English?

Reviewer #1: (No Response)

Reviewer #2: Yes

6. Review Comments to the Author

Reviewer #1: (No Response)

Reviewer #2: I believe the authors have adequately addressed my comments and questions. I have also reviewed their responses to other reviewers' critiques and have no further questions.

7. PLOS authors have the option to publish the peer review history of their article (what does this mean?). If published, this will include your full peer review and any attached files.

Reviewer #1: No

Reviewer #2: **Yes: **Stephen R. Walsh

---

## [Editor Report · Acceptance letter]

3 Feb 2023

PONE-D-22-28685R1 

Comparison of three dosing intervals for the primary vaccination of the SARS-CoV-2 mRNA Vaccine (BNT162b2) on magnitude, neutralization capacity and durability of the humoral immune response in health care workers: a prospective cohort study. 

Dear Dr. Duong:

I'm pleased to inform you that your manuscript has been deemed suitable for publication in PLOS ONE. Congratulations! Your manuscript is now with our production department. 

Kind regards, 

on behalf of

Dr. Glenda Canderan 

Academic Editor

PLOS ONE